# Biocompatible Catanionic Vesicles from Arginine-Based Surfactants: A New Strategy to Tune the Antimicrobial Activity and Cytotoxicity of Vesicular Systems

**DOI:** 10.3390/pharmaceutics12090857

**Published:** 2020-09-09

**Authors:** Aurora Pinazo, Ramon Pons, Ana Marqués, Maribel Farfan, Anderson da Silva, Lourdes Perez

**Affiliations:** 1Department of Surfactant and Nanobiotecnnology, Institute of Advanced Chemistry of Catalonia, IQAC-CSIC, c/Jordi Girona, 18-26, 08034 Barcelona, Spain; aurora.pinazo@iqac.csic.es (A.P.); ramon.pons@iqac.csic.es (R.P.); anderson.ramos@iqac.csic.es (A.d.S.); 2Department of Biology, Healthcare and the Environment, Section Microbiology, Faculty of Pharmacy, University of Barcelona, Av. Joan XXIII s/n, 08028 Barcelona, Spain; ammarques@ub.edu (A.M.); mfarfan@ub.edu (M.F.)

**Keywords:** catanionic vesicles, amino acids, biocompatible surfactants, antimicrobial activity, antibiofilm activity, hemolysis, bacterial selectivity

## Abstract

Their stability and low cost make catanionic vesicles suitable for application as drug delivery systems. In this work we prepared catanionic vesicles using biocompatible surfactants: two cationic arginine-based surfactants (the monocatenary *N*α-lauroyl-arginine methyl ester—LAM and the gemini *N*α,*N*ϖ-bis(*N*α-lauroylarginine) α, ϖ-propylendiamide—C_3_(CA)_2_) and three anionic amphiphiles (the single chain sodium dodecanoate, sodium myristate, and the double chain 8-SH). The critical aggregation concentration, colloidal stability, size, and charge density of these systems were comprehensively studied for the first time. These catanionic vesicles, which form spontaneously after mixing two aqueous solutions of oppositely charged surfactants, exhibited a monodisperse population of medium-size aggregates and good stability. The antimicrobial and hemolytic activity of the vesicles can be modulated by changing the cationic/anionic surfactant ratio. Vesicles with a positive charge efficiently killed Gram-negative and Gram-positive bacteria as well as yeasts; the antibacterial activity declined with the decrease of the cationic charge density. The catanionic systems also effectively eradicated MRSA (Methicillin-resistant *Staphylococcus Aureus*) and *Pseudomonas aeruginosa* biofilms. Interestingly, the incorporation of cholesterol in the catanionic mixtures improved the stability of these colloidal systems and considerably reduced their cytotoxicity without affecting their antimicrobial activity. Additionally, these catanionic vesicles showed good DNA affinity. Their antimicrobial efficiency and low hemolytic activity render these catanionic vesicles very promising candidates for biomedical applications.

## 1. Introduction

The versatile vesicular nanosystems have attracted considerable attention due to their promising therapeutic applications. They may encapsulate polar drugs in their aqueous compartment and hydrophobic compounds can be incorporated into the hydrocarbon domain [1]. The most widely studied are liposomes [2], biocompatible and biodegradable spherical aggregates composed of natural polar lipids, which present no significant cytotoxicity issues. [3,4,5]. Nevertheless, liposomes also have some important drawbacks: (a) their raw materials, usually phospholipids, are costly; (b) their production can require a high energy input and lengthy preparation methods with multiple steps (sonication, extrusion, homogenization) [6], (c) these vesicles are typically non-equilibrium structures with limited colloidal stability [7] and (d) natural lipids can easily undergo chemical degradation by hydrolysis.

Catanionic vesicles [8], which spontaneously self-assemble without any energy requirements after the mixing of inexpensive cationic and anionic surfactant solutions [9], have been investigated as potential alternatives to liposomes. Among their advantages, catanionic vesicles can be thermodynamically or kinetically stable for long periods of time, they can be prepared using simple and economic surfactants, and their size, surface density, flexibility, and permeability can be tailored by adjusting the concentration, cationic/anionic mixing ratio, temperature, and chain length of the components and the addition of salts or co-solvents [8]. Due to their physicochemical and biological properties, catanionic vesicles are of special interest for applications in nanotechnology and biomedicine. [10,11,12,13,14].

In most studies on catanionic mixtures, alkyl ammonium bromide surfactants are chosen as the positively charged components [15]. Recently, due to their enhanced properties, catanionic mixtures containing gemini surfactants such as bis-quaternary ammonium salts have also been used. However, these cationic surfactants are not biodegradable, and their cytotoxicity precludes their use in some biomedical applications. In this regard, the conjugation of non-toxic molecules is being explored as a promising approach to produce biocompatible and biodegradable surfactants that can be incorporated in catanionic mixtures [16,17,18,19,20,21].

The arginine amino acid is also an excellent raw material to prepare cationic surfactants with antimicrobial activity. Years ago, our group addressed the synthesis and physico-chemical characterization of long-chain *N*α-acyl arginine methyl ester compounds and arginine-based gemini surfactants [17]. *N*α-acyl arginine methyl esters are cationic surfactants with high biodegradability, moderate hemolytic activity, and a surface activity comparable to that of conventional long-chain quaternary ammonium salts. Moreover, these compounds showed good antimicrobial and antifungal activity, the most active agent being the 12-carbon homologue (LAM, Figure 1). Moreover, LAM combined with anionic surfactants, formed mixtures containing polydisperse catanionic vesicles coexisting with other kinds of structures, [22].

The gemini surfactants consisted of two symmetrical long-chain residues of C_12_ (*N*α,*N*ω-bis(*N*α-lauroylarginine)α,ω-alkylendiamides [C_n_(LA)_2_ series] and C_10_ (*N*α,*N*ω-bis(*N*α-caproylarginine)α, ω-alkylendiamides [C_n_(CA)_2_ series] linked by amide bonds to an α,ω-alkylenediamine spacer chain of varying length (*n* = 2–10). The arginine-based gemini surfactants showed critical micellar concentration (cmc) values [17] one or two orders of magnitude lower than the corresponding monomeric surfactants and were much more efficient in lowering surface tension. These gemini surfactants also exhibited excellent biodegradation and good antimicrobial activity. Interestingly, the most active antimicrobial gemini surfactant, C_3_(CA)_2_, performed better than LAM, the most active monocatenary surfactant [17]. Taking into account the biological and physico-chemical properties of single-chain and arginine-based gemini surfactants, their use in the preparation of catanionic mixtures may be advantageous for several biomedical applications. Moreover, it has been reported that the mode of action of cationic amphiphiles against bacteria hampers the development of antimicrobial resistance, one of the most serious threats to public health [23] that calls for the development of new antimicrobial systems.

Here we prepared catanionic mixtures simply by mixing cationic arginine-based surfactants with anionic surfactants (sodium laurate, sodium myristate and 8-SH) (Figure 1). Three different systems were studied: one composed of two single-chain surfactants, LAM and sodium myristate, a second prepared by mixing the cationic gemini C_3_(CA)_2_ with the single chain sodium laurate, and a third composed of C_3_(CA)_2_ and the double chain 8-SH. Systems containing gemini surfactants were also prepared with different contents of cholesterol. The critical aggregation concentration as well as the size and charge density of the resulting aggregates as a function of surfactant mixing ratio were determined by fluorescence, conductivity, dynamic light scattering, and zeta potential studies. Antimicrobial and antibiofilm activity of all the catanionic mixtures was tested against four Gram-positive bacteria, three Gram negative bacteria, and the yeast-like pathogenic fungus *Candida albicans*. The antibiofilm activity against two problematic bacteria (MRSA and *Pseudomonas aeruginosa*) was determined. The interaction of the catanionic mixtures with human cells was evaluated using erythrocytes. Additionally, the DNA-binding capacity of these systems was assessed by ethidium bromide exclusion experiments.

The study had three main aims: first, to prepare catanionic mixtures using green biocompatible surfactants, second, to design new formulations with high antimicrobial activities and moderate toxicity and third, to shed light on how the mixing ratio, the number of alkyl chains in the ion pairs and the nature of polar heads affect the biological and physico-chemical properties of catanionic vesicles. We expect that the findings described in this work will contribute to the understanding of the biological activity of catanionic mixtures from amino acid-based surfactants and help to rationalize the design of new and safe antimicrobial formulations.

## 2. Materials and Methods

### 2.1. Materials

All solvents were reagent grade and used without further purification. Deuterated solvents were purchased from Eurotop (Cambridge, UK). Mueller-Hinton Broth was purchased from Difco Laboratories (Detroit, MI, USA). Water from a Milli-Q Millipore system (Millipore, Burlington, MA, USA) was used to prepare aqueous solutions. Sodium dodecanoate and myristate were from Sigma (San Luis, MO, USA), while the other surfactants used to prepare the catanionic mixtures (LAM, C_3_(CA)_2_ and 8-SH) were prepared in our laboratory following procedures already described [24,25]. Briefly, the gemini surfactant C_3_(CA)_2_ was obtained with a purity of 99% by chemical condensation of the previously protected single *N*α-decyl-l-arginine to the propanediamine in presence of an activating agent. A final deprotection reaction was carried out to obtain the target surfactant. LAM was synthesized by the condensation of the lauroyl chloride (Sigma, San Luis, MO, USA) to the amino group of the commercial arginine methyl ester in basic medium (Sigma, San Luis, MO, USA). Sodium 8-hexadecylsulfate (8-SH) was prepared by sulfonation of 8-hexadecanol (Sigma, San Luis, MO, USA) with chlorosulfonic acid in acetic acid (Sigma, San Luis, MO, USA). The resulting sulfonic acid was neutralized by sodium bicarbonate (Sigma-aldrich, San Luis, MO, USA). The detailed synthetic procedure as well as the characterization of pure compounds (checked by high performance liquid chromatography (HPLC), Mass spectrometry (MS), proton and carbon-13 nuclear magnetic resonance (^1^HNMR, ^13^CNMR), and elemental analysis) are provided elsewhere.

### 2.2. Preparations of Catanionic Mixtures

Catanionic formulations were prepared by mixing the appropriate amount of aqueous solutions of pure surfactants (Table 1). As cholesterol is a very hydrophobic lipid not soluble in water, cholesterol-containing vesicles were prepared using a hydration method. Firstly, solutions of anionic surfactants, cationic surfactants, and cholesterol (Sigma-aldrich, San Luis, MO, USA) were prepared in methanol (Honeywell Riedel-de Haën, Seelze, Germany). Then 2 mL of mixtures containing the corresponding amount of every pure solution were prepared and the solvent was evaporated. The resulting film was then hydrated with 2 mL of water and sonicated at 50 °C for 15 min.

### 2.3. Fluorescence Measurements

A Shidmadzu RF 540 spectrofluorometer (Shimadzu, Kioto, Japan) was used to measure CMC by fluorescence, using pyrene (Sigma-aldrich, San Luis, MO, USA) as a fluorescence probe at 25 °C. From a stock solution of catanionic mixtures (1 mM) prepared in a pyrene aqueous solution of 10^−6^ M, different dilutions were prepared. The fluorescence emission spectra of these dilutions were recorded from 340 to 450 nm after excitation at 332 nm. The ratio of the first to the third vibrionic peaks (*I*_I_/*I*_III_) in the emission spectra of pyrene depends on the environment polarity and can be used to calculate the cmc or cac.

### 2.4. Conductivity

Conductivity of surfactant solutions was measured using an Orion Cond Cell 011010A (Thermo scientific, Waltham, MA, USA) with platinized platinum electrodes in conjunction with a Thermo Orion 550A with a cell constant of 0.998 cm^−1^. The cell constant was calibrated with NaCl/KCl solutions of known conductivities. Samples for conductivity measurements were prepared in Millipore ultra-pure water. Measurements were made at increasing concentrations to reduce errors from possible electrode contamination. The conductivity of water was subtracted from the measured conductivity of each sample.

### 2.5. NMR Measurements

The ^1^HNMR spectra were acquired with a Varian 400 mHz (Varian, Palo Alto, CA, USA) using a simple one-pulse experiment with a spectral width of 10,000 Hz and a pre-acquisition delay of 5 s. Binary mixtures were prepared in 1 mL of D_2_O (1 mM for the catanionic mixtures and 5 mM for LAM and sodium myristate) of at least 99.8% atom D%. The water impurity in the D_2_O was used as an internal reference.

### 2.6. ς-Potential and Size Distribution Analysis

The ς-potential of the formulations was measured with Laser Doppler electrophoretic mobility measurements using a Zetasizer 2000 (Malvern Instruments Ltd., Malvern, UK), at 25 °C. All analyses were done in triplicate. The ς-potential values and standard deviations were acquired directly from the instrument. The size distribution of all formulations was determined by using the dynamic light scattering technique (DLS). The DLS unit determining vesicle size was a Malvern Zeta Nanosizer (Malvern, Malvern, UK), working at 632.8 nm and 25 °C. The DLS experiment started 5 min after the sample solutions were placed in the DLS optical system to allow the sample to equilibrate at the selected temperature. The scattering intensity was measured at a 173° angle to the incident beam. The backscatter detection allows measurements in turbid dispersions, minimizing multiple scattering effects. Depending on the system turbidity, the unit automatically determines the sample thickness under investigation and focuses the beam at a given position from the cell walls. At least 10 runs were performed for each sample. The viscosity value (0.8872 mPa×s) and the refractive index (1.33) of water were used for all the measurements. The correlation function is used by the system to calculate the size distribution. The software supplied by the manufacturer (Zetasizer v6.20, Malvern, Malvern, UK) automatically determines the optimum parameters for the CONTIN73 algorithm for monomodal as well as multimodal distributions in order to produce a size distribution profile.

### 2.7. Antimicrobial Activity

Antimicrobial activity was assessed in vitro by determining the minimum inhibitory concentration (MIC) values using the broth microdilution method [26]. Antimicrobial assays were carried out using 4 Gram positive bacteria, 3 Gram negative bacteria and one fungal strain: methicillin resistant *Staphylococcus aureus* ATCC 43300 (MRSA), *Bacillus subtilis* ATCC 6633, *Kocuria rhizophila* 9341, *Staphylococcus epidermidis* ATCC 12228, *Klebsiella pneumoniae* ATCC 4532, *Pseudomonas aeruginosa* ATCC 27853, *Escherichia coli* ATCC 8739, *Candida albicans* ATCC 10231 (American Type Culture Collection, Manassas, VA, USA). Frozen stocks of bacterial isolates were spread on Muller Hinton agar plates and incubated overnight at 37 °C. Then, a bacterial suspension of 0.5 McFarland turbidity was prepared in Muller Hinton broth with a final pH of 7.3 ± 0.1 at 25 °C. To obtain the inoculum, these suspensions were diluted 10 times in Muller Hinton broth to give 10^7^ colony forming units (CFU)/mL. Two-fold dilutions of pure surfactants and their catanionic mixtures in the same broth were prepared to obtain a final concentration range of 4 to 250 μM in the 96 microtiter plates. 10 μL of nutrient broth culture of each bacterial strain was then added to achieve a final microorganism density of ca. 5 × 10^5^ CFU/mL. Every assay was performed in triplicate. The lowest concentration of the catanionic mixtures that inhibits visible growth of every microorganism (after 24 h of incubation at 37 °C) was regarded as the MIC. The minimum bactericidal concentration (MBC) was recorded as the lowest concentration, killing 99.9% of the bacterial inoculum after 24 h incubation at 37 °C. MBC values were determined by inoculating in nutrient agar plates 100 μL of bacterial suspension from the wells whose concentrations are ≥ MIC. Plates were incubated at 37 °C for 24 h. Each test was repeated at least three times.

### 2.8. Antibiofilm Activity

Bacteria were grown overnight in Tryptic-Soy agar at 37 °C for 24 h. The bacteria were suspended in lysogeny broth with glucose (1%) at 1.5 × 10^8^ CFU/mL. For mature biofilm formation, 200 μL of the diluted bacterial suspension was added to each well in a 96-well microplate and incubated at 37 °C for 24 h. The wells were gently rinsed with phosphate buffered saline (PBS) and 200 μL of the corresponding catanionic formulation was added to each well and incubated again at 37 °C for 24 h. Two-fold product concentration ranges were used (125–2.5 μM). After treatment, the biofilms were fixed with 100 μL methanol for 15 min. After the plates were air dried, the biofilms were stained with 50 μL of 0.1 of % crystal violet for 25 min. The plates were rinsed with water and 150 μL of 96% ethanol was added to all 96 wells to dissolve the remaining crystal violet stained biofilm. The O.D value was recorded at 570/630 nm. Controls were performed with non-catanionic formulations (untreated). Each assay was carried out four times and the results were averaged.

### 2.9. Hemolysis Assay

For the preparation of erythrocyte suspensions, fresh blood was taken from rabbit using a procedure approved by the institutional ethics committee on animal experimentation. The erythrocytes were washed three times in PBS (pH 7.4). Finally, erythrocytes were suspended in PBS at a cell density of 8 × 10^9^ cells/mL.

To determine the hemolytic activity of the catanionic mixtures prepared in this work, we adapted the procedure described by Pape et al. [27]. A series of different volumes of a concentrated sample, ranging from 10 to 80 μL, were placed in small Eppendorf tubes containing 25 μL of erythrocyte suspension and PBS was added to each tube to a total volume of 1 mL. Samples were shaken for 10 min at room temperature and the tubes were then centrifuged (5 min at 10,000 rpm). The percentage of hemolysis was determined by comparing the absorbance (540 nm) of the supernatant of the samples with that of the control totally hemolysed with distilled water. Concentration–response curves were determined from the hemolysis results and used to calculate the concentration inducing 50% hemolysis (*HC_50_*).

### 2.10. Ethidium Bromide Fluorescence

The fluorescence spectra of ethidium bromide (EB) (3 mL, 0.16 μM) (Sigma-aldrich, San Luis, MO, USA) were determined with a PTI Fluorescence Master System fluorescence spectrophotometer (Horiba, Kyoto, Japan). The excitation wavelength (λ_exc_) was 490 nm and the emission range 530–700 nm. 2 μg of DNA from salmon (average MW 1.3 × 10^6^ ca. 2000 base pairs) was then added to 3 mL of EB solution and fluorescence spectra were recorded again under the same conditions. Increasing volumes of pure surfactants or catanionic mixtures were then incorporated to the EB/DNA solution and fluorescence spectra were recorded after each addition. The EB release from the DNA/EB complex due to the interaction of the nucleotide with the catanionic vesicles was calculated using the following equation
$$\%\;\mathrm{EB}=\;\frac{I_{0}-I}{I_{0}-\;I_{\mathrm{EB}}}\;\times \;100$$
here *I*_0_ and *I*_EB_ are the maximum fluorescence intensities obtained for the EB and EB/DNA solutions and *I* is the highest fluorescence intensity achieved with different catanionic mixture concentrations.

## 3. Results and Discussion

### 3.1. Critical Aggregation Concentration

In this work we used 2 cationic amino acid-based surfactants and 3 anionic surfactants (Figure 1) to prepare a range of catanionic mixtures at room temperature. The surfactant mixtures contained different proportions of: (a) LAM and sodium myristate, both with a single chain (LM), (b) double-chain cationic gemini C_3_(CA)_2_ and single-chain anionic sodium laurate (C3L) and (c) C_3_(CA)_2_ and anionic 8 S-H, both with a double chain (C3S) (Figure 2). Spontaneous vesicle formation by binary mixtures of oppositely charged surfactants depends on both the composition of the formulation and the surfactant structure. Usually a single phase of catanionic vesicles is formed in the diluted cation- or anion-rich region of the surfactant [28]. Thus, in this work each mixture was prepared in 3 different proportions: one rich in the cationic surfactant, one rich in the anionic surfactant, and an equimolecular formulation. Each system was prepared at two different total concentrations (1 and 5 mM).

The critical aggregation concentration (cac) of catanionic systems and the critical micellar aggregation (cmc) of the pure surfactants were determined by fluorescence and conductivity. In general, cac corresponds to a more generic concept than cmc, it is used when the aggregates in equilibrium with the monomers are not micelles.

Fluorescence spectra of pyrene in different concentrations of surfactant solutions are shown in Figure 3. For all systems, the *I*_I_/*I*_III_ ratio decreased as the surfactant concentration increased. At low surfactant concentration, the *I*_I_/*I*_III_ ratio remained constant at about 1.6, indicating that the probe was located in a hydrophilic environment. At higher concentrations, the ratio decreased drastically, in response to the hydrophobic domains of micelles or other aggregates being formed by the surfactant molecules. The cac or cmc can be obtained from the intersection of these two sections in the curves (Table 2). Conductivity measurements of LAM and LM mixtures are shown in Figure 3a, the conductivity values increased linearly with the surfactant concentration until reaching a breakpoint corresponding to the cmc or cac (Table 2). At higher concentrations, further conductivity increases were more gradual. Below the cmc, the conductivity is due to both free counterions and charged surfactant monomers. Above the cmc, the effective micelle charge is reduced due to the binding of some counterions to the micelle surface. Moreover, the ionic mobility of micelles is lower than that of monomers.

The cmc values of the pure surfactants were found to be in good agreement with the literature values [25,29]. Due to their more hydrophobic character, the cmc of the cationic gemini surfactant C_3_(CA)_2_ and the double chain surfactant 8-SH was lower than that of LAM and sodium laurate.

The cac values of the mixed surfactant systems were lower than the cmc of individual cationic and anionic surfactants, likely due to the synergistic effects of strong electrostatic interactions and the formation of large aggregates such as vesicles [30,31]. The spontaneous formation of vesicles requires an optimum value of the packing parameter *P*, which is defined by three geometric structural parameters [32]: *P* = *V*/*l*_c_*a*_0_, where *V* is the volume of the hydrophobic portion, *l*_c_ is the length of the hydrophobic group, and *a*_0_ is the head group area of the surfactants. Micelles are formed when the *P* value is less than ½, whereas higher values promote the formation of vesicles and lamellar phases. When two oppositely charged single-chain surfactants are mixed, the head groups interact electrostatically to behave as a single ion pair, which acts as a pseudo double-tailed zwitterionic surfactant. These strong electrostatic interactions reduce the effective head group area of the catanionic ion pair and increase the area of the hydrophobic group per molecule relative to that of each single-chain surfactant. As a result, the *P* value is close to 1, which favors the formation of vesicles at very low concentrations.

For the catanionic mixtures prepared with single-chain surfactants (LM), the cac decreased from 4.7 and 1.6 mM for the pure compounds to 0.3 and 0.4 mM for 8.2LM and 2.8LM, respectively. The 5.5LM mixture precipitated at an equimolar concentration, a frequent occurrence in equimolecular mixtures of oppositely charged surfactants due to the complete neutralization of ionic charges [15]. The decrease in cac was higher for the C3L mixtures, dropping from 3 and 9 mM for the pure compounds to 0.08–0.3 for the catanionic mixtures, and even more so for the C3S mixtures. This behavior can be ascribed to the different catanionic ion pairs formed in each mixture (Figure 2). In the case of LM, the ion pairs contain only two alkyl chains. In contrast, the C3L mixtures allow the formation of pseudo surfactants of 3 and 4 hydrophobic chains, as cationic gemini surfactants contain two alkyl chains and two positive charges (Figure 2), and the C3S mixtures result in pseudo surfactants with four or six alkyl chains. These types of ion pairs could explain the low cac values obtained for these systems. It was also observed that the cac values of C3L and C3S increased with the proportion of C_3_(CA)_2_. As the cationic surfactants have two cationic charges and the anionic only one, 8.2C3L contained 14 free cationic charges for every 16, and 2.8C3L had 4 free anionic charges for every 8. The anion-rich system, being less ionic, therefore has lower cac values. This would also explain the absence of precipitation in the 5.5 samples, as neutralization would occur at a 2/1 ratio.

The cac was also determined by conductivity measurements [33]. Using this technique, it was only possible to determine the cmc of LAM and the cac of the LM mixtures (Figure 3a) and the values were slightly lower than those obtained by fluorescence (Table 2). For the gemini mixtures, there was not an obvious inflection point in the conductivity curves, probably due to the low cmc values and low counterion binding in catanionic mixtures, where a significant part of the counterion binding role is already assumed by the minority surfactant in the mixture.

The behavior of the new compounds in aqueous solution was also studied by ^1^HNMR (Figure 4). ^1^HNMR spectra were recorded using a Varian 400 MHz to monitor self-aggregation of pure surfactants as well as their mixtures. In the ^1^HNMR spectra of pure LAM and sodium myristate (Figure 4a), the stereochemical splitting resonance corresponding to the different CH_2_ and CH_3_ groups of the molecule is clearly resolved, and the typical Lorentzian-shaped spectra corresponding to the presence of classical micelles can be observed. The ^1^HNMR spectra of the catanionic mixtures showed considerable signal-broadening, which may be associated with the slow motion of the surfactants due to the formation of large, stable aggregates [34,35]. The ^1^HNMR spectra also indicated the formation of catanionic vesicles. Similar behavior was observed for the C3S (Figure 4b) and C3L mixtures (figures not shown).

### 3.2. Size Distribution and ς-Potential

The cac values as well as the NMR measurements of the binary mixtures suggested the spontaneous formation of catanionic vesicles. To confirm their formation, the nature of the aggregates was studied. The hydrodynamic diameters as well as the size distribution of the aggregates were determined by dynamic light scattering (DLS) studies (Figure 5, Table 3 and Table 4). The solution appearance was also analysed, as a change in the turbidity of a surfactant solution can indicate alterations in the amount and/or size of the aggregates. Table 3 summarizes the hydrodynamic diameters and the polydispersity index (PdI) obtained from the DLS measurements at day 0. The table also shows the visual appearance of every formulation.

The DLS results corresponding to the LM mixtures indicate that catanionic vesicles were spontaneously formed after mixing the two monocatenary surfactant aqueous solutions in a ratio of 8/2 and 2/8. The PdI values indicate that both 8.2LM and 2.8LM contained a monodisperse population of small vesicles with a mean *D_h_* of 121 nm and 190 nm, respectively. The DLS results were supported by visual observations, as the 8.2LM and 2.8LM dispersions were bluish in color, which is typical of catanionic vesicles sized in the visible wavelength range (Table 3, Figure 5a) [25]. At an equimolar ratio, the mixed solution became very turbid and the surfactant mixture was prone to precipitate. When all positive charges are neutralized by the negative ones, the hydrophobic content of the system is very high, leading to phase separation. Precipitation when the surfactant ratio nears equimolarity has been observed for several catanionic mixtures [15] and is attributed to the formation of zwitterionic ion pairs without a positive or negative net charge.

Table 3 also contains the values of the ς-potential, a parameter commonly used to assess the stability of catanionic vesicles under storage [36]. The aggregates tend to repel each other if they have large positive or negative ς-potential values (+30 mV or −30 mV), otherwise the mixtures start to flocculate and precipitate. The storage stability also depends on the size distribution and is generally improved by high attractive inter-particle interactions (average particle diameter < 500 nm), a *PdI* < 0.4. The 8.2 and 2.8 LM formulations exhibited large ς-potential values (−65 and 64 mV) and a small average diameter, although phase separation occurred in both formulations after 72 h.

In all the mixtures of the gemini surfactant C_3_(CA)_2_ with 8-SH (C3S formulations), the translucent bluish color observed with the naked eye also indicated the spontaneous assembly of aggregates with sizes comparable to the wavelength of light, such as catanionic vesicles (Table 3). For these systems, small vesicles with a mean diameter of 138–183 nm were observed and the *PdI* was below 0.3, indicating a narrow vesicle size range [37] (Figure 5d). The ς-potential depends on the cationic/anionic ratio of the mixtures: those that were cation-rich gave a large positive value of +40.4 mV and when anion-rich, a large negative value. The equimolecular mixture also showed a positive value (+15.3 mV); as the gemini cationic surfactant possesses two positive charges, at this ratio only half of the cationic charges are neutralized by the anionic surfactant. There was a clear correlation between the excess charges and the ς-potential of the mixtures: for every 10 molecules, the 2.8 mixture had 4 negative net charges, the equimolecular mixture 5 positive charges and the 8.2 mixture 14 positive charges. The size distribution and ς-potential values suggest a very good and durable stability. Accordingly, visual inspection of the formulation after 20 days indicated no phase separation and the DLS measurements showed only a small increase in the mean diameter of the aggregates, the *PdI* remaining similar. The size distribution and stability of the three C3S mixtures were also measured at a higher concentration (5 mM). A less transparent bluish solution was observed, attributable to a higher aggregate concentration, as the values of the other parameters (*D_h_*, *PdI*, and ς-potential) were largely unchanged. After 20 days, these mixtures remained stable, with only a slight increase in size and polydispersity (Figure 5e, Appendix A).

Mixing the gemini surfactant C_3_(CA)_2_ with sodium laurate (C3L formulations) yielded mostly milky solutions with large aggregates, with a *D_h_* of around 500 nm for the 8.2 and 2.8 formulations and 900 for the equimolecular mixture (Figure 5b,c, Table 3). The aggregates were found to have a higher particle size and *PdI* compared to the C3S vesicles. As expected, the ς-potential of the cation-rich and equimolecular mixtures was positive and negative, respectively, for the anion-rich mixture. However, the anionic C3L vesicles had a lower anionic charge compared to those of the C3S formulations. The high ratio of sodium laurate decreased the ς-potential values but to a lesser extent than 8-SH. Being a stronger base than 8-SH, this anionic surfactant is less negatively charged and its interactions with the gemini surfactant gave rise to large, more insoluble aggregates. This result could be ascribed to the pH sensitivity of sodium laurate, which has a pK_a_ of 5.4, close to the pH of pure water, and was therefore likely to be present in the formulations in ionized and neutral forms. The presence of non-ionized sodium laurate molecules could also explain the large size of the aggregates, given the absence of attractive electrostatic interaction between these species and the cationic molecules. At 5 mM, the 5.5C3L and 8.2C3L mixtures did not show the typical appearance of catanionic mixtures, the viscosity of the solutions suggesting the presence of different aggregates (Table 3). The C3L mixtures were poorly stable under storage and most of them suffered phase separation after 2 weeks (Appendix A). In general, these formulations exhibited weak intra-particle interactions (related to the large size) and weak repulsive inter-particle interaction (related to their small ς-potential), which leads to low stability. Similar results were found by Want el al., who studied mixtures of a bis-quaternary ammonium surfactant with different anionic amphiphiles. For bisQuat/sodium laurate, phase separation occurred over a wide range of mixing ratios, while the mixtures with sodium lauroyl sulphate did not precipitate in any of the formulations studied [38].

It has been reported that the introduction of additives such as cholesterol or alcohols into vesicular bilayers can modify their physico-chemical and biological properties [39,40]. For example, cholesterol can influence membrane permeability. This hydrophobic molecule can eliminate the gel/liquid crystalline phase transition of lipid bilayers, inducing an ordering effect at temperatures above the *Tm* (phase transition temperature) and a disordering effect at temperatures below the Tm [41].

Taking into account these considerations, we decided to investigate how the incorporation of cholesterol in the catanionic mixtures would affect their physico-chemical and biological properties. Table 4 shows the *D_h_*, *PdI*, ς-potential and visual appearance of C3S and C3L (1 mM) prepared with 10% or 20% cholesterol. The addition of 10% cholesterol to C3S solutions did not alter their visual appearance, which remained bluish. The DLS measurements showed that this additive generally promoted the formation of mixtures with a single population of larger vesicles (mean *D_h_* 118–463 nm) and higher polydispersity (*PdI* 0.225–0.390). The ς-potential values of the cationic systems were largely unchanged, while the anionic aggregates exhibited lower values. A similar effect was produced by 20% cholesterol: the C3S mixtures maintained good stability, with no phase separation observed in any sample after 30 days, and the size and polydispersity values remained similar (Appendix A).

In the C3L mixtures, the addition of cholesterol at both 10% and 20% had a strong influence on vesicular size, the mean *D_h_* values being generally lower than for formulations without the additive (Table 4). Moreover, larger ς-potential values were obtained for the anionic mixtures. The most notable difference produced by cholesterol in these formulations was in their storage stability. After 30 days the visual appearance was maintained, and no phase separation was observed (Appendix A). The DLS values after 7 and 30 days showed that the mean *D_h_* and polydispersity were largely unchanged. The improved stability of these mixtures, compared with solutions without cholesterol, can be ascribed to their lower aggregate size and higher ς-potential values. A previous study also found that cholesterol enhanced the physical stability of catanionic vesicles prepared with mixtures of monocatenary and dicatenary ammonium-based surfactants and sodium dodecyl sulphate [42]. Once incorporated into the vesicular bilayers, the cholesterol sterol ring tends to maximize contact with neighboring hydrocarbon chains, thus improving the bilayer mechanical strength. Moreover, by increasing the distance between the charged head groups in the vesicles, the addition of cholesterol would reduce counterion binding and consequently increase the ionic strength of the vesicles.

### 3.3. Antimicrobial Activity

In the last ten years antibiotic resistance has grown to alarming levels [43] due to the abuse or misuse of antibiotics [44]. While antibiotic resistance is difficult and perhaps impossible to overcome, the development of new biocompatible antimicrobial systems with novel mechanisms of action can be an interesting strategy to minimize its emergence.

In this study, we investigated the antimicrobial activity of novel catanionic mixtures composed of biodegradable surfactants. The cationic surfactant C_3_(CA)_2_ was selected for its antimicrobial properties, its specific spectrum of action and its mechanism of interaction with bacteria [45]. Moreover, previous studies indicate that the gemini C_3_(CA)_2_ exhibits higher antimicrobial activity than the monocatenary LAM. On the other hand, lipoamino acids are also particularly attractive as antiviral agents and certain acyl amino acid derivatives have been found to inhibit influenza neuraminidase [46].

The antimicrobial activity of the catanionic vesicle dispersions was evaluated against a panel of clinically relevant microorganisms by determining the minimum inhibitory concentration (MIC) and the minimum bactericidal concentration (MBC). The MIC and MBC values corresponding to the catanionic mixtures and their individual components are shown in Figure 6 and Appendix A.

While the cationic gemini surfactant exhibited good antimicrobial activity against the three types of microorganisms tested, the anionic surfactants did not affect their growth. As the bacterial cytoplasmic membrane carries a net negative charge, a positive charge and suitable hydrophobic/hydrophilic character is essential for the antimicrobial potential of surfactants [47,48]. The activity of C_3_(CA)_2_ was similar against the three Gram-positive bacteria (MIC values of 9 μM), being slightly lower against the Gram-negative *P. aeruginosa* and *E. coli* (MIC 36 μM), and without any effect on *K. pneumoniae* at the highest concentration tested. The tolerance of Gram-negative bacteria to cationic amphiphiles is well known, being due to the intrinsic permeability barrier of the outer membrane and the role of lipopolysaccharides [49].

In general, the antimicrobial activity of the pure C_3_(CA)_2_ was moderately higher compared to the catanionic mixtures, although the latter were still very effective. This behavior could be ascribed to the aggregate morphology of the systems as well as to the cationic charge density. A pure gemini surfactant tends to form spherical micelles, whereas catanionic mixtures contain large aggregates. Additionally, all cationic charges in pure systems are free, but in the mixtures, some are neutralized by the anionic surfactant. It has been reported that the self-assembly properties (aggregate size, morphology and stability) affect the antimicrobial activity of surfactant systems [50], after electrostatic targeting the bacterial surface, small aggregates can easily insert into and disrupt the lipidic bacterial membrane via hydrophobic interactions [51,52].

Our results demonstrate that the surfactant antimicrobial activity mainly depends on the anionic/cationic ratio. The effectivity of the 8.2 cationic/anionic systems was similar to that of pure C_3_(CA)_2_, decreasing with the proportion of the cationic surfactant. Changes in the surfactant ratio altered the most important factors affecting the antimicrobial activity: the hydrophobicity and cationic density of the colloidal systems. The outermost surface of the bacterial cytoplasmic membrane carries a net negative charge, which leads to a high binding affinity with catanionic vesicles with large positive ς-potential values. This phenomenon was clearly observed in the 2.8 formulations, whose negative ς-potential values led to a drastically reduced antimicrobial activity.

C3L formulations were the most active against bacteria and yeast. For example, 2.8C3L was effective against almost all the tested microorganisms, whereas 2.8C3S showed no activity against any of them. This difference in performance can be ascribed to the structural stability and dynamics of the aggregates. The efficient interaction of surfactant monomers with antibacterial membranes is favored when the molecular aggregates are relatively unstable and easily disassemble into monomers. The physico-chemical studies of these systems indicate that more weakly interaction occurred between C_3_(CA)_2_ and sodium laurate due to the pH-sensitive behavior of this anionic surfactant. It can then be assumed that the ion pair formed in the C3L mixtures was more readily separated. Recent reports indicate that cellular internalization is strongly dependent on aggregate stability; the more stable aggregates enter cells by endocytosis, disassembling inside, whereas aggregates that can easily dissociate intercalate into the membranes as monomers, penetrating cells directly [53,54].

Interestingly, the pure C_3_CA_2_ did not suppress growth of *K. pneumoniae*, unlike the C3L mixtures. It is also remarkable that all catanionic formulations were effective against MRSA. The WHO, based on various criteria, has created a global priority list of antibiotic-resistant pathogens. The situation is particularly critical for infections associated with hospitalization, which are mainly caused by the Gram-negative *K. pneumoniae* and *P. aeruginosa* as well as by the Gram-positive *Staphylococcus aureus*.

Cationic amphiphiles exert antimicrobial activity mainly by disintegrating bacterial membranes via electrostatic and hydrophobic interactions. In the case of C_3_(CA)_2_, flow cytometry analysis and electron micrographs indicate that *E. coli* cells treated with this surfactant are killed by membrane permeabilization [45]. It is thus expected that the catanionic mixtures share this antibacterial mode of action. As bacteria find it difficult to circumvent such a nonspecific mode of action as membrane-disruption, such antimicrobial systems are of great interest for reducing the growth of bacterial resistance [55,56].

The incorporation of 10% cholesterol to the catanionic mixtures did not modify their antimicrobial activity, which also decreased as the anionic surfactant percentage grew (Figure 6, Appendix A). Formulations with 20% cholesterol exhibited lower activity, possibly because of an increased hydrophobicity, due to the highly apolar nature of the additive (Appendix A).

The action of an antibacterial agent on a bacterial strain can also be characterized by the minimum bactericidal concentration (MBC), the dosage at which >99.9% of bacteria are killed. This parameter was determined for the cholesterol-containing catanionic mixtures (Appendix A), and the MBC/MIC ratio ranged from 1 to 2, indicating a bactericidal effect against the tested strains.

The results suggest that the catanionic vesicles obtained by simply mixing an arginine-based gemini surfactant with an anionic surfactant retain the antimicrobial activity of the cationic surfactant. Moreover, the antimicrobial potency of these systems can be easily modulated by changing the anionic surfactant or the cationic/anionic ratios.

### 3.4. Antibiofilm Activity

Most bacteria form biofilms when they grow on surfaces. MRSA and *P. aeruginosa* are currently two problematic bacteria with the ability to develop robust biofilms that are 10 to 1000 times more resistant to conventional antimicrobials than planktonic–state bacteria. In this context, it is crucial that new antimicrobial systems can also disperse bacterial biofilms. Accordingly, pre-established mature biofilms of MRSA and *P. aeruginosa* were treated with the novel colloidal formulations and their eradication was measured by a crystal violet staining assay (Figure 7).

Both C_3_(CA)_2_ and the catanionic mixtures had a significant antibiofilm effect against the two strains tested. Up to 70% of MRSA biofilms were dispersed by C_3_(CA)_2_ at 36 μM, and up to 60% by the C3S mixture at 72 μM. The C3L mixtures were less effective, reducing MRSA biofilms by 50% at 125 μM and having no dispersal affect below this concentration. While low concentrations (18 μM) of C_3_(CA)_2_ dispersed about 60% of *P. aeruginosa* biofilms, the C3S mixture performed even better, removing up to 70% at subMIC (36 μM). In accordance with these results, it was recently reported that a biosurfactant loaded in phosphatidylcholine liposomes showed higher activity against MRSA biofilm than when free [57].

The antibiofilm activity of these catanionic formulations can be ascribed to both their cationic character and their alkyl chains. A proposed mechanism for biofilm eradication by cationic amphiphiles first involves perturbation of the extracellular polymeric substance through electrostatic interactions, which disperses the biofilm and leads to the death of the exposed planktonic cells [58].

The effects of amino acid-based surfactants on biofilms have been scarcely reported to date, and to our knowledge, this is the first study on the antibiofilm activity of vesicles prepared with these surfactants. Tack-seung Kim et al. [59] recently found that lauroyl arginine ethyl ester effectively removed biofilm from reverse osmosis membranes. In general, cationic surfactants are more efficient at inhibiting the development of biofilms than dispersing them [57,59]. The efficiency of the studied catanionic systems in eradicating MRSA biofilms is similar to that reported for quaternary ammonium surfactants [58]. This good activity and the green chemical procedures used for their preparation make them attractive alternatives to existing methods of biofilm eradication.

### 3.5. Hemolytic Activity

Many catanionic mixtures have been designed as drug delivery systems, but their biomedical application is severely limited by strong cytotoxicity. As it is crucial that antimicrobial formulations are not toxic for mammalian cells, the toxicity of the novel catanionic mixtures toward human cells was assessed using erythrocytes, one of the most widely used cell membrane systems to study surfactant–membrane interactions [27].

The percentage of hemolysis plotted against concentration is shown in Figure 8 and the curves were used to calculate the *HC_50_* (the concentration that causes 50% hemolysis) (Table 5).

As shown in Table 5 and Figure 8, sodium laurate did not exhibit hemolytic activity at the highest concentration tested (200 μM). The cationic C_3_(CA)_2_ as well as the anionic 8-SH showed low hemolytic activity, the latter being less toxic. The low interaction of sodium laurate with the erythrocytes can be explained by its low hydrophobicity (cmc = 9 mM) and its pH-sensitivity. The gemini C_3_(CA)_2_ and 8-SH have similar hydrophobicity, although the higher cationic charge density of the former allowed a more efficient interaction with the cell membrane. The therapeutic index (*TI* = *HC_50_*/*MIC*) correlates the *MICs* with the *HC_50_* to reveal the selectivity of the mixtures against bacteria. Notably, the *MIC* values of C_3_(CA)_2_ were generally lower than the *HC_50_* (*TI* > 1), indicating selective action against the bacterial membrane (Figure 9, Appendix A).

The catanionic mixtures prepared with this gemini surfactant showed lower hemolytic character (Figure 8). The hemolytic activity of these formulations seems to be governed by the cationic charge; in both cases the formulations with the highest percentage of cationic gemini surfactants exhibited lower *HC_50_* values. C3L had a notably low hemolytic nature, its high *HC_50_* values resulting in a higher TI compared to C_3_(CA)_2_, indicating an improvement in selectivity (Figure 9, Appendix A). The enhanced selectivity of the C3L vesicles is associated with their lower cationic charge density (and smaller ς-potential), which was sufficient to interact strongly with the surface of the bacterial membrane but not with the limited negative charge of erythrocytes.

The incorporation of cholesterol in these formulations considerably increased the *HC_50_* values (Table 5), especially at 20%. The selectivity of the C3S cholesterol formulations improved substantially (Appendix A, Figure 9). The enhanced selectivity and safety obtained by introducing cholesterol can be due to three factors: a) cholesterol is an apolar compound that modifies the hydrophobic/hydrophilic balance of the formulations, b) this additive decreases the cationic charge density and the aggregate size, and c) a moderate fraction of cholesterol in the catanionic vesicles increases their biocompatibility because it resembles the erythrocyte membrane [60].

The selectivity of C3L cholesterol formulations did not increase. Nevertheless, these mixtures can still be very useful for certain biomedical applications as the vesicles are cationically charged and have very little cellular toxicity.

Catanionic vesicles possess a tremendous potential for use in different biomedical applications. However, these systems are leaky and have safety issues due to their possible cytotoxicity. This work indicates that the main factors affecting the antimicrobial activity and cytotoxicity of catanionic vesicles (hydrophobicity, cationic charge, and morphology) can be easily tuned by changing the anionic/cationic surfactant ratio and the surfactants used, and by adding an appropriate additive. In this regard, the preparation of catanionic mixtures using a cationic gemini arginine-based surfactant can be a key design principle to endow antimicrobial systems with lower cytotoxicity while retaining or even enhancing their efficiency. Additionally, the obtained results could be very useful for the optimization and development of safe and efficient antimicrobial systems that hamper the emergence of resistance. This hypothesis is confirmed by previous studies that report good selectivity of catanionic mixtures prepared with C_3_(CA)_2_ and an anionic biosurfactant against Gram-negative and Gram-positive bacteria [61]. The data obtained here also shed light on the structure–activity relationships underlying the toxicity of these mixtures, which need to be understood for their transfer from the bench to the bedside.

### 3.6. DNA Binding Properties

Catanionic mixtures prepared with biocompatible surfactants could be suitable for the development of vesicles as DNA carriers. With the aim of exploring the potential of the surfactant mixtures studied in this work for such applications, their interaction with DNA was preliminarily studied using ethidium bromide (EB) intercalation experiments.

Appendix A shows the emission fluorescence spectra of EB with increasing concentrations of cationic surfactants and their catanionic vesicles. As expected, the intensity of EB fluorescence increased strongly after the ethidium ion intercalated between the DNA base pairs. The addition of catanionic mixtures causes a quenching of the fluorescence emission intensity (Appendix A). These results indicate that catanionic aggregates compacted DNA and EB was gradually released from the EB-DNA complex.

Figure 10 shows the percentage of EB displaced from the EB/DNA complex as a function of the concentration of pure cationic surfactants or the catanionic mixtures. The pure monocatenary LAM possesses good DNA binding capacity and positively charged LAM catanionic vesicles also compacted DNA but with a lower effectivity. Previous studies report that catanionic vesicles prepared by mixing a monocatenary arginine-based surfactant bearing a C12 alkyl chain with sodium octyl and cetyl surfactants were also able to compact DNA. [62,63]. The DNA affinity of LAM and the 8.2LM formulation was similar to that of monocatenary surfactants with long alkyl chains [64,65]. In this context, it has been reported that arginine-based lipids also exhibit DNA-binding properties through the hydrogen bonds of the guanidinium group [66].

The percentage of EB displaced by the gemini C_3_(CA)_2_ and its catanionic vesicles indicates an excellent DNA binding capacity. Formulations with the arginine-based gemini surfactant were far more efficient at compacting the DNA than those prepared with LAM.

The higher DNA affinity shown by the gemini-based systems could be ascribed to two structural factors: the presence of two cationic charges situated on the protonated guanidine groups, which improves the electrostatic interactions with the negatively charged phosphate of DNA, and the existence of two alkyl chains, which increases the hydrophobic character of the formulations, reducing the amount of cationic surfactant needed to saturate the oligonucleotides [67,68]. These results are in accordance with those of gemini pyridinium [68] and histidine-based gemini surfactants [69].

The DNA affinity of catanionic mixtures was slightly lower than that of the pure surfactants. Notwithstanding this, the percentage of EB displaced by these catanionic vesicles from the EB/DNA complex indicates an excellent DNA binding capacity. The lower activity of the catanionic formulation can be explained by their lower cationic density.

The present results indicate a strong interaction between gemini-based catanionic vesicles and DNA. These results are important from the toxicological point of view, as the catanionic vesicles, especially the 5.5 formulations, exhibited much less hemolytic activity while retaining almost all their capacity to compact DNA. This quality endows these systems with an added advantage for biological applications and suggests that C3L catanionic mixtures are promising candidates for further investigation in gene delivery.

## 4. Conclusions

In this work, the physico-chemical and biological properties of catanionic vesicles prepared with biocompatible surfactants are described in detail. Spontaneously formed vesicles were obtained by mixing aqueous solutions of cationic arginine-based surfactants with anionic surfactants. These catanionic mixtures displayed a very low critical aggregation concentration with respect to the individual components, indicating a strong attractive interaction between the oppositely charged amphiphilic molecules, which form highly hydrophobic ion pairs. The mixtures containing cationic arginine-based surfactants provided excellent monodisperse populations of medium-sized catanionic vesicles that remained reasonably stable for at least 30 days.

The net surface charge as well as the antimicrobial and hemolytic activity of the vesicles can be tuned by altering the surfactant mixing ratio, which allows the development of systems with selective activity against bacteria and without toxicity against red blood cells. Interestingly, the selectivity and stability of the catanionic vesicles were further improved by the incorporation of cholesterol. It was also demonstrated that the novel surfactant systems can effectively disperse MRSA and *P. aeruginosa* biofilms.

The results of this work indicate that designing catanionic vesicles from biocompatible surfactants is a useful strategy in the search for selective antimicrobial systems with an optimal balance between antimicrobial activity and cytotoxicity. The low cytotoxicity and high stability of the vesicles are also promising features for the development of new drug delivery systems for systemic applications. Moreover, given their properties, the vesicles could simultaneously act as a vehicle and drug.

Overall, the data obtained provide novel insights into the effect of catanionic vesicles on bacteria and cellular systems, which could contribute to developing their practical application in biomedical fields.

## Figures and Tables

**Figure 1 pharmaceutics-12-00857-f001:**
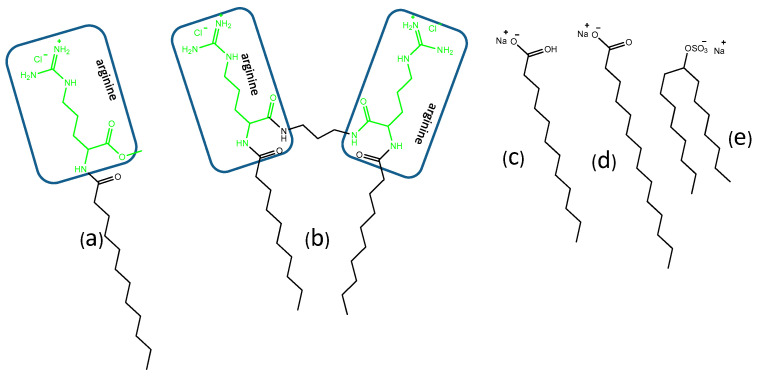
Chemical structure of the cationic and anionic surfactants used in this work: (**a**) LAM (**b**) C_3_(CA)_2_, (**c**) Sodium laurate, (**d**) Sodium myristate, and (**e**) 8-SH.

**Figure 2 pharmaceutics-12-00857-f002:**
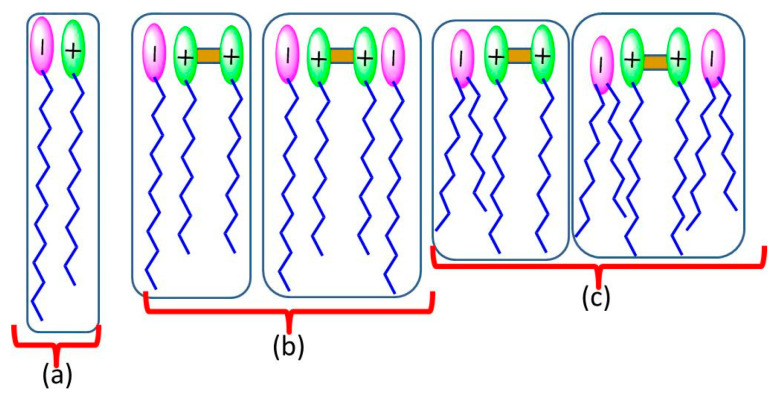
Ion pairs types in the prepared binary mixtures. (**a**) pseudo-dialkyl surfactants in LM systems (**b**) pseudo-trialkyl and tetraalkyl surfactants in C3L mixtures and (**c**) pseudo-tetraalkyl and hexaalkyl surfactants in C3S systems.

**Figure 3 pharmaceutics-12-00857-f003:**
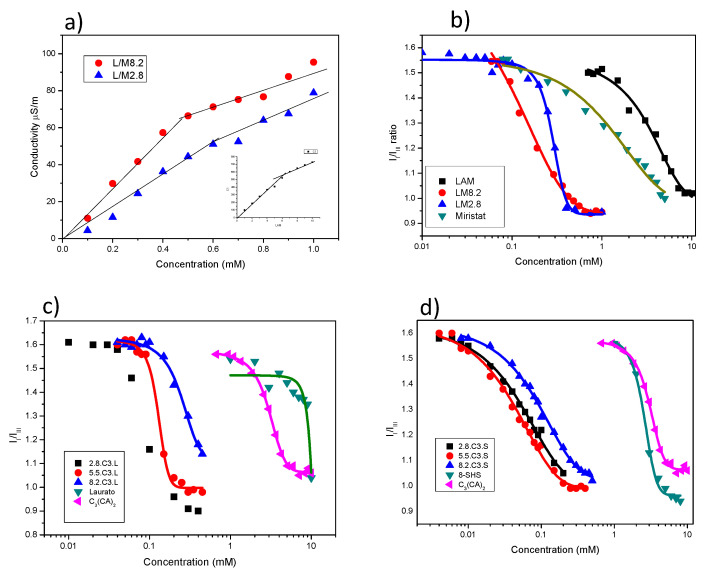
(**a**) Conductivity against concentration for LAM and LM mixtures, (**b**) Fluorescence measurements of LAM and LM mixtures, (**c**) Fluorescence measurements of C_3_(CA)_2_, sodium laurate and their corresponding catanionic mixtures, and (**d**) Fluorescence spectra against concentration for C_3_(CA)_2_, 8-SH and their catanionic mixtures.

**Figure 4 pharmaceutics-12-00857-f004:**
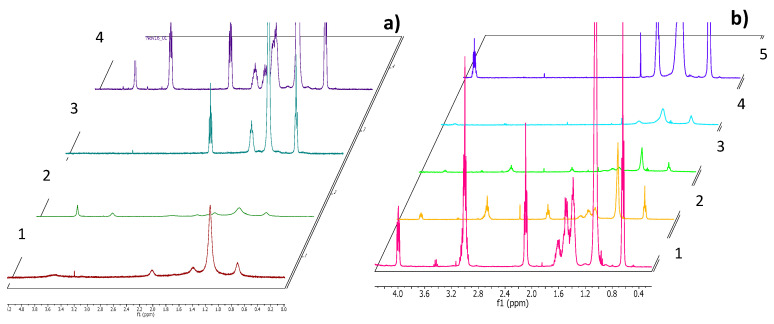
^1^HNMR spectra of (**a**) LAM 6 mM (4), Myristate 3 Mm (3), 2.8LM 1 mM (2) and 8.2LM 1 mM (1) (**b**) C3S mixtures (**1** C_3_(CA)_2_ 5 mM, **2** 8.2 C3S 1 mM, **3** 5.5 C3S 1 mM, **4** 2.8 C3S 1 mM, **5** 8-SH 5 mM).

**Figure 5 pharmaceutics-12-00857-f005:**
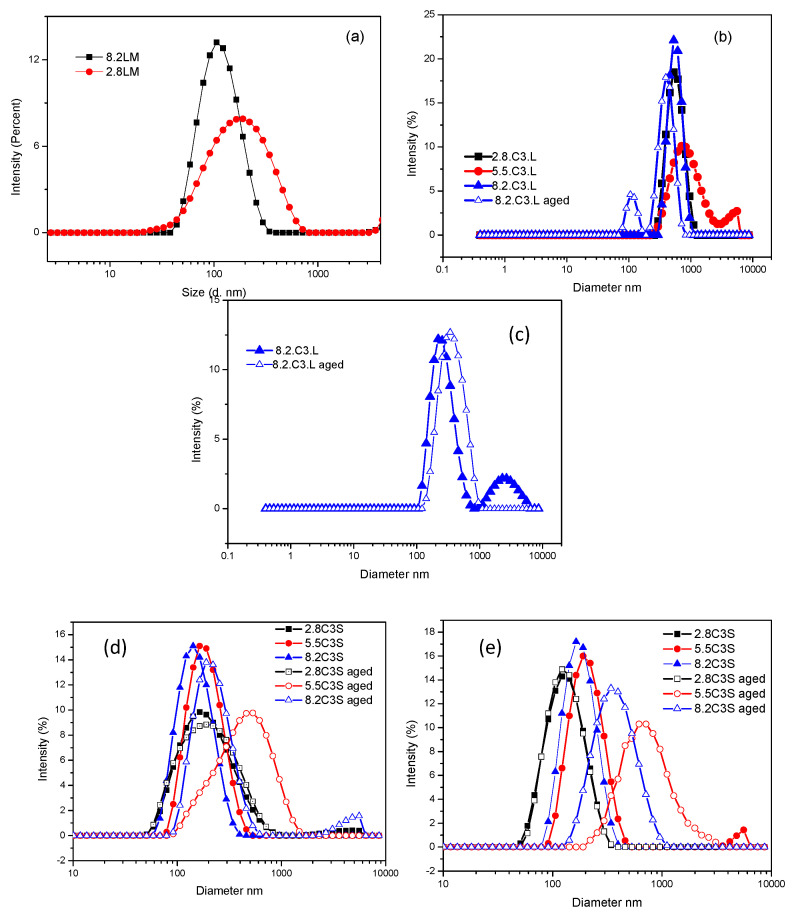
Size distribution (diameter Dh) for the binary mixtures of (**a**) LM (**b**) C3L 1 mM day 0 and day 20 (**c**) C3L 5 mM day 0 and day 20, (**d**) C3S 1 mM day 0 and day 20, and (**e**) C3S 5 mM day 0 and day 20.

**Figure 6 pharmaceutics-12-00857-f006:**
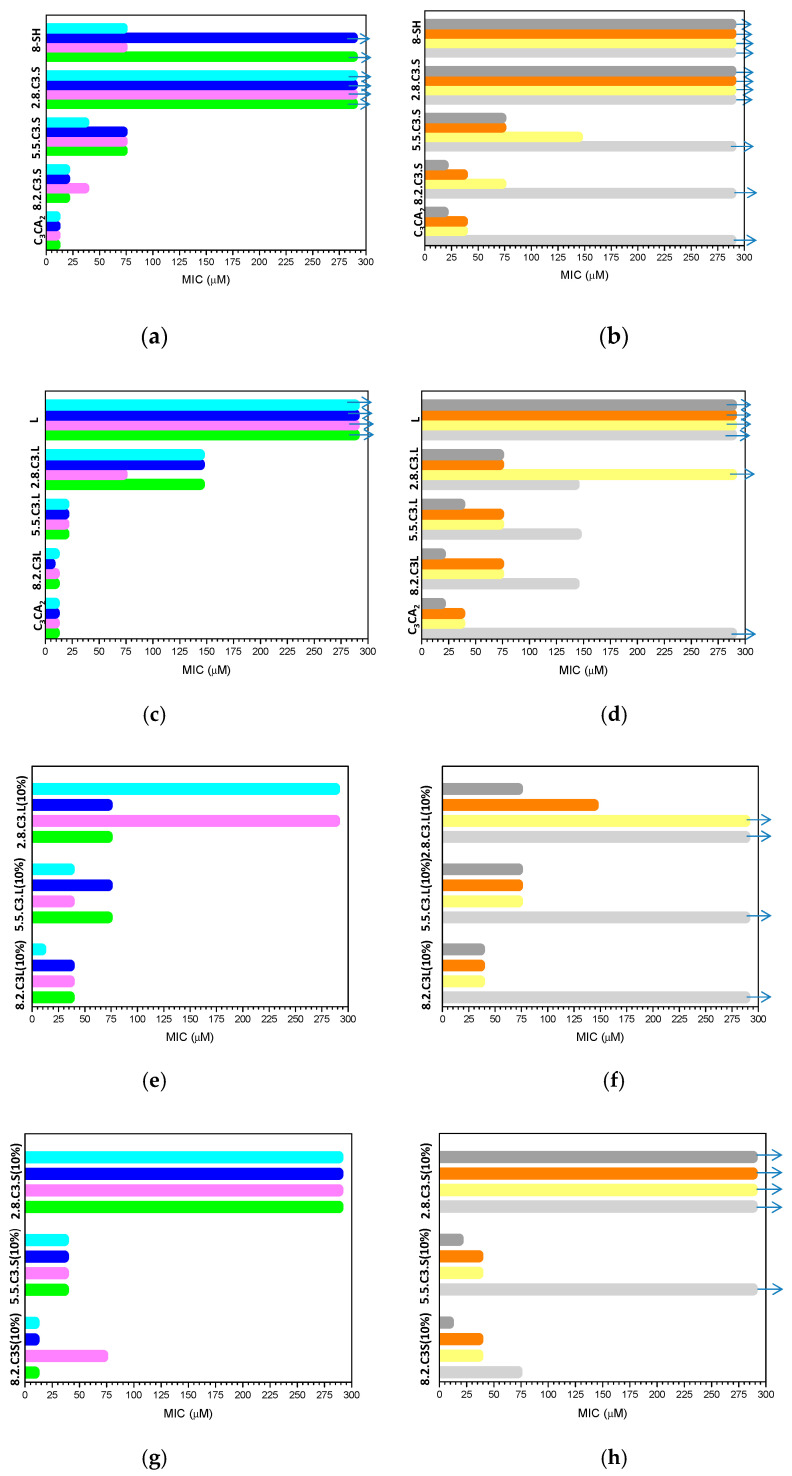
*MIC* (μM) of pure surfactants as well as their catanionic mixtures without and with (10%) cholesterol against Gram-positive and Gram-negative bacteria: (**a**) C3S mixtures against Gram-positive bacteria (**b**) C3S mixtures against Gram-negative bacteria, (**c**) C3L mixtures against Gram-positive bacteria, (**d**) C3L mixtures against Gram-negative bacteria, (**e**) C3L(10%) mixtures against Gram-positive bacteria, (**f**) C3L mixtures against Gram-negative bacteria, (**g**) C3S (10%) mixtures against Gram-positive bacteria and (**h**) C3S (10%) mixtures against Gram-negative bacteria ■ MRSA ■
*B. subtilis*
■
*K. rhizophila*
■
*S. epidermidis*
■
*K. pneumoniae*
■
*P. aeruginosa*
■
*E. coli*
■
*C. albicans*. Arrows indicates that the highest concentration tested did not inhibit the growth of the microorganism.

**Figure 7 pharmaceutics-12-00857-f007:**
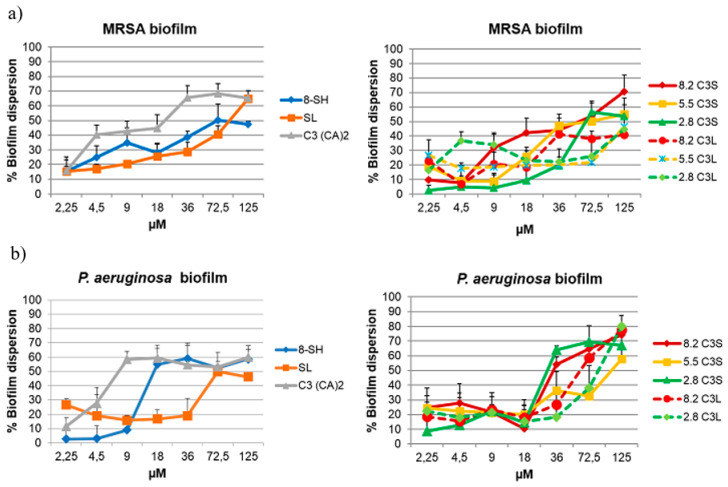
(**a**) MRSA biofilm dispersion (%) against concentration (μM) of pure surfactants and their catanionic mixtures, (**b**) *P. aeruginosa* biofilm dispersion (%) against concentration (μM) of pure surfactants and their catanionic mixtures.

**Figure 8 pharmaceutics-12-00857-f008:**
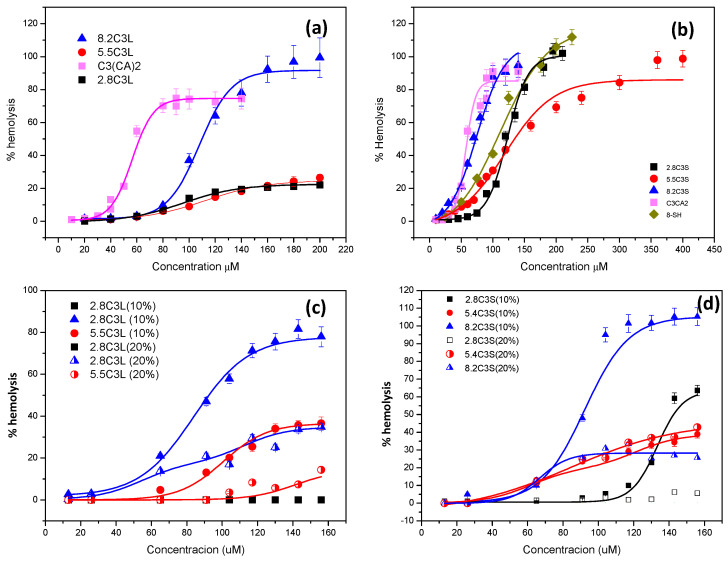
Hemolysis percentage against surfactant concentration for the C3L (**a**) and C3S (**b**) systems and their cholesterol containing mixtures (**c**,**d**).

**Figure 9 pharmaceutics-12-00857-f009:**
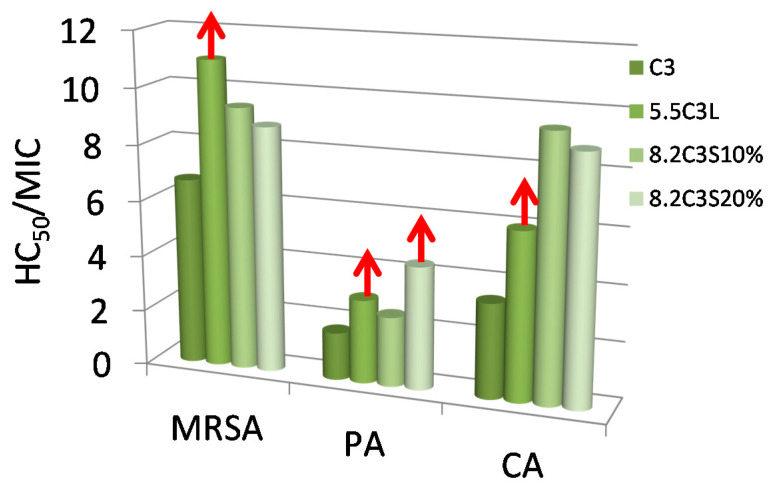
Therapeutic Index (*HC_50_*/*MIC*) of C_3_CA_2_ and some of their catanionic mixtures against a Gram-positive bacteria (MRSA), a Gram-negative (*P. aeruginosa* (PA)) and a yeast (*C. albicans* (CA)). Arrows indicate that the *TI* is greater than or equal to the value in the figure.

**Figure 10 pharmaceutics-12-00857-f010:**
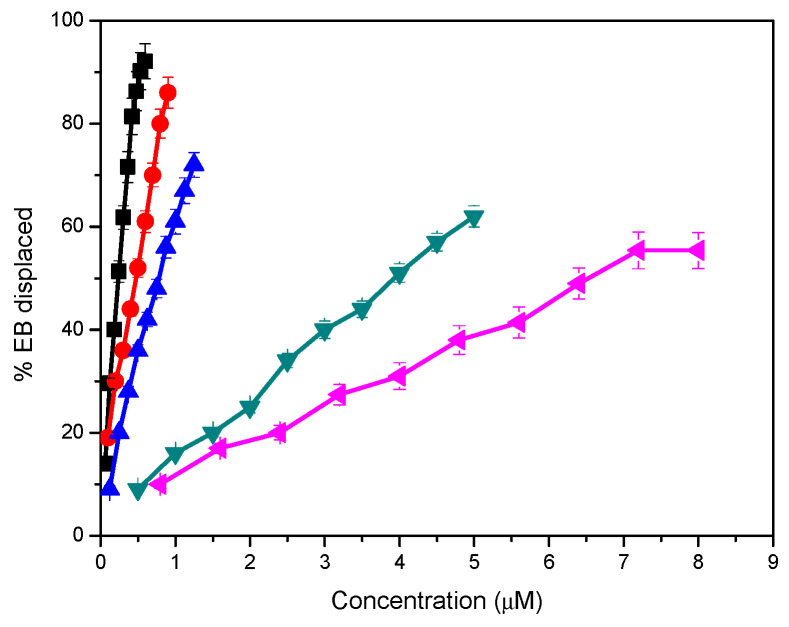
Release of ethidium bromide (EB) (%) from the EB/DNA complex against concentration of cationic surfactants and their catanionic vesicles (μM): C_3_(CA)_2_ (■) 8.2 C3L (●), 5.5 C3L (▲) LAM (◄), 8.2 LM (►).

**Table 1 pharmaceutics-12-00857-t001:** Acronyms and composition (molar) of the catanionic mixtures studied in this work.

**Catanionic Mixtures**									
**Acronym**	**LAM (%)**		**Sodium Myristate (%)**	**Acronym**	**C_3_(CA)_2_ (%)**	**8SH (%)**	**Acronym**	**C_3_(CA)_2_ (%)**	**Sodium Laurate (%)**
8.2.LM	80		20	2.8.C3S	20	80	2.8.C3L	20	80
2.8.LM	20		80	5.5.C3S	50	50	5.5.C3L	50	50
				8.2.C3S	80	20	8.2.C3L	80	20
**Cholesterol Containing Catanionic Mixtures**									
Acronym		**Cholesterol (%)**		**C_3_(CA)_2_ (%)**	**8SH (%)**	**Acronym**	**Cholesterol (%)**	**C_3_(CA)_2_ (%)**	**Sodium Laurate (%)**
2.8.C3S(10%)		10		18	72	2.8.C3L (10%)	10	18	72
2.8.C3S(20%)		20		16	64	2.8.C3L (20%)	20	16	64
5.5.C3S(10%)		10		45	45	5.5.C3L (10%)	10	45	45
5.5.C3S(20%)		20		40	40	5.5.C3L (20%)	20	40	40
8.2.C3S(10%)		10		72	18	8.2.C3L (10%)	10	72	18
8.2.C3S(20%)		20		64	16	8.2.C3L (20%)	20	64	16

**Table 2 pharmaceutics-12-00857-t002:** Critical aggregation concentration (cac) of pure surfactants and their catanionic mixtures.

Single Chain Surfactants			Double Chain Surfactants	
System	cac (mM) Fluorescente	cac (mM) Conductivity	System	cac (mM) Fluorescence
LAM	4.7	6.2	C_3_(CA)_2_	3.2
Sodium myristate	1.6	-	8-SH	2.5
8.2 LM	0.3	0.45	Sodium laurate	9.1
2.8 LM	0.4	0.6	2.8C3S	0.064
			5.5C3S	0.048
			8.2C3S	0.105
			2.8C3L	0.08
			5.5C3L	0.13
			8.2C3L	0.28

**Table 3 pharmaceutics-12-00857-t003:** Mean diameter, polydispersity index (*PdI*), ς-potential values, and visual appearance of the catanionic mixtures at day 0. Photograph of the LM mixtures, left 8.2LM and right 2.8LM. Photographs corresponding to the C3L and C3S systems: left 2.8, center 5.5, and right 8.2.

Concentration	Formulation	Size *D_h_*(nm)	*PdI*	ς-Potential (mV)	Visual Aspect
**1 mM**	2.8 LM	190	0.400	−65.7	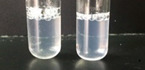
	5.5 LM	Precipitation			
	8.2 LM	121	0.198	+64.7	
	2.8 C3S	138	0.109	−33.4	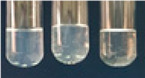
	5.5 C3S	212	0.180	+15.3	
	8.2 C3S	183	0.090	+40.4	
	2.8C3L	581	0.133	−5.4	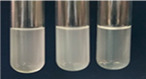
	5.5C3L	924	0.360	+27.3	
	8.2C3L	572	0.320	+37.4	
**5 mM**	2.8 C3S	217	0.223	−52.5	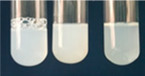
	5.5 C3S	194	0.151	+36.4	
	8.2 C3S	156	0.174	+61.5	
	2.8C3L	272	0.262	−20	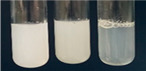
	5.5C3L		GEL		
	8.2C3L		GEL		

**Table 4 pharmaceutics-12-00857-t004:** Mean diameter, polydispersity index (*PdI*), ς-potential values, and visual appearance of the catanionic mixtures with cholesterol (1 mM) at day 0. Photographs corresponding to the C3L and C3S systems: left 8.2, center 5.5, and right 2.8.

Cholesterol	Formulation	Size *D_h_*(nm)	PdI	ς-Potential (mV)	Visual Aspect
**10%**	2.8C3L	566	0.175	−15	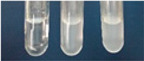
	5.5C3L	218	0.202	40	
	8.2C3L	58	0.400	44	
	2.8 C3S	242	0.390	−17	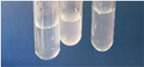
	5.5 C3S	463	0.343	17	
	8.2 C3S	118	0.225	40	
**20%**	2.8C3L	251	0.216	−12	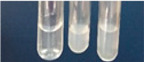
	5.5C3L	191	0.179	24	
	8.2C3L	207	0.335	29	
	2.8 C3S	255	0.410	−10	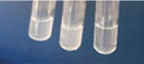
	5.5 C3S	272	0.332	16	
	8.2 C3S	218	0.447	39	

**Table 5 pharmaceutics-12-00857-t005:** *HC_50_* values of pure surfactants and their catanionic mixtures.

Pure Compounds	*HC_50_* μM	Catanionic Mixtures	*HC_50_* μM	10%Chol Mixtures	*HC_50_* μM	20% Chol Mixtures	*HC_50_* μM
C_3_(CA)_2_	60 ± 3.1	2.8C3S	120 ± 6	2.8C3S (10%)	>156	2.8C3S (20%)	>156
8-SH	100 ± 4.2	5.5.C3S	160 ± 12	5.5.C3S (10%)	>156	5.5.C3S (20%)	>156
SL	>200	8.2.C3S	79 ± 5	8.2.C3S (10%)	85 ± 5	8.2.C3S (20%)	>156
		2.8C3L	>200	2.8C3L (10%)	>156	2.8C3L (20%)	>156
		5.5.C3L	>200	5.5.C3L (10%)	>156	5.5.C3L (20%)	>156
		8.2.C3L	110 ± 11	8.2.C3L (10%)	100 ± 8	8.2.C3L (20%)	>156

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
