# Peer review of "Biocompatible Catanionic Vesicles from Arginine-Based Surfactants: A New Strategy to Tune the Antimicrobial Activity and Cytotoxicity of Vesicular Systems"

_pharmaceutics, 2020, doi:10.3390/pharmaceutics12090857_

Round 1

Reviewer 1 Report

Authors in this manuscript studied the physico-chemical and biological properties of catanionic vesicles prepared with biocompatible surfactants.

The authors have done very detailed and systematic research on cationic vesicles. The manuscript is richly illustrated, which makes it very easy to follow the exposed material.

Some parts of the manuscript may seem too long, such as an introduction.

In the part of studying the mixture of surfactants, it is necessary to clarify:

What is the difference between the critical aggregation concentration (cac) and the critical micellar concentration (cmc)?

For the binary mixtures of surfactants without cholesterol why not use cmc instead od cac? You can use some of thermodynamic model (regular solution theory, Rodenas function, etc.) for thermodynamic (stabilization) characterization of binary surfactant systems. In the formed vesicle what is the molar ratio of surfactant, becoues this is different compared of the molar ration in the initial surfactants binary mixture.

 Whether there is a difference between binary mixed micelles and cationic vesicles, please explain.

Author Response

Thank you very much for your constructive evaluation, which has helped us improve this manuscript. Below you will find our answers (in red) to your concerns (in black).

Comments and Suggestions for Authors

Authors in this manuscript were studied the physico-chemical and biological properties of catanionic vesicles prepared with biocompatible surfactants.The authors have done very detailed and systematic research on cationic vesicles. The manuscript is richly illustrated, which makes it very easy to follow the exposed material.

Some parts of the manuscript may seem too long, such as an introduction.

The introduction and some part of the results and discussion section have been reduced.

In the part of studying the mixture of surfactants, it is necessary to clarify:

What is the difference between the critical aggregation concentration (cac) and the critical micellar concentration (cmc)?

cac is used in the literature both when the aggregates in equilibrium with the monomers are not micelles and when this is not proved but probable. That is, cac corresponds to a more generic concept than cmc. This sentence has been introduced in the revised manuscript (Line 286)

For the binary mixtures of surfactants without cholesterol why not use cmc instead od cac? You can use some of thermodynamic model (regular solution theory, Rodenas function, etc.) for thermodynamic (stabilization) characterization of binary surfactant systems. In the formed vesicle what is the molar ratio of surfactant, becoues this is different compared of the molar ration in the initial surfactants binary mixture.

It is not clear whether catanionic mixtures have a true cmc (that is, equilibrium between micelles and monomers) but, rather equilibrium between monomers and vesicles or lamellae fragments can be envisaged because the solutions show bluish tones from very low concentrations. Because of this we prefer the use of cac in this context.

Concerning the thermodynamic models, as far as we know, a true development of thermodynamic models for the mixed systems with 2:1 contrary charged systems is not available. By applying directly Rodenas or other thermodynamic models, the calculated composition of the mixed systems tends to equimolar in any case while what would be expected is to approach the equivalent point, which is 0.33:0.66. It is not clear whether those models can be applied taking into account that there is a third compound in the system, that is, the catanionic compound.

 Whether there is a difference between binary mixed micelles and cationic vesicles, please explain.

Binary mixed micelles are micelles formed by two different surfactants while cationic vesicles are vesicles (spherical aggregates containing one or more surfactant bilayers) which have a net positive charge; the  V-potential value is higher than 0. Cationic vesicles can be prepared by a cationic surfactant or by different surfactant mixtures: cationic/anionic systems or cationic/non ionic systems.

Reviewer 2 Report

The authors discussed the biophysical properties of the catanionic vesicles composed of arginine-based cationic surfactants and anionic surfactants. Vesicle formation was monitored by a number of physico-chemical and biomedical methods for various vesicles compositions and surfactant structures. The obtained results are very valuable in view of their potential application in pharmacy and medicine.

The manuscript is well written. All conclusions are supported by data. The figures are carefully prepared allowing the reader to have a clear picture of all aspects discussed. The experimental methods applied for the characterization of vesicles are adequate for reaching the main goals of the paper.

Some minor points:

line 267: is “such as the the micelle”, please correct it,

line 299: is “0.2 and 0.3 mM for 8.2LM and 2.8LM”, but according to Table 2 it should be “0.3 and 0.4 mM for 8.2LM and 2.8LM”,

line 303: the value of 0.28 in Table 2 should be rounded to 0.3, not to 0.2

The paper is a very interesting contribution and should be published in Pharmaceutics.

Author Response

Thank you very much for your constructive evaluation, which has helped us improve this manuscript. Below you will find our answers (in red) to your concerns (in black).

The authors discussed the biophysical properties of the catanionic vesicles composed of arginine-based cationic surfactants and anionic surfactants. Vesicle formation was monitored by a number of physico-chemical and biomedical methods for various vesicles compositions and surfactant structures. The obtained results are very valuable in view of their potential application in pharmacy and medicine. The manuscript is well written. All conclusions are supported by data. The figures are carefully prepared allowing the reader to have a clear picture of all aspects discussed. The experimental methods applied for the characterization of vesicles are adequate for reaching the main goals of the paper.

Some minor points:

line 267: is “such as the the micelle”, please correct it,

The sentence has been corrected

line 299: is “0.2 and 0.3 mM for 8.2LM and 2.8LM”, but according to Table 2 it should be “0.3 and 0.4 mM for 8.2LM and 2.8LM”,

Fixed (Line 342)

line 303: the value of 0.28 in Table 2 should be rounded to 0.3, not to 0.2

Fixed (Table 2, Line 342))

The paper is a very interesting contribution and should be published in Pharmaceutics.

Reviewer 3 Report

The paper was very carefully written. The background information and result description were very comprehensive, although most of the figure legends require more information to aid understanding.

I am a little confused about the aim of the study. According to the introduction, the purpose of generating the catanionic vesicles was for drug carrier and delivery. However, a large part of the experiments were focused on the characterisation of the antibacterial properties of the materials. It is not clear if the author also intended to use the catanionic vesicles as an alternative of the antibiotic medication. If so, will the vesicles be used as a type of antimicrobial medicine and/or sterilization solution? It needs some discussions to clarify the potential applications of the materials. The aim of the study needs to be much more focused.

The results and discussions are all mixed up, which makes the section very lengthy. It is very difficult to pinpoint the actual results as they are embedded in the discussions. It would be beneficial to separate the discussion from the result to highlight the actual results.

Minors: 

  • Line 92, please check the “es” in “…value [72] es one or…” is correct.
  • In Figure 1, (d) came before (c). It’s better to change their position.
  • Figure 4 needs to label A) and B)
  • The legends of Figure 8, 9, 10, and13 need much more comprehensive.

Author Response

Thank you very much for your constructive evaluation, which has helped us improve this manuscript. Below you will find our answers (in red) to your concerns (in black).

Comments and Suggestions for Authors

The paper was very carefully written. The background information and result description were very comprehensive, although most of the figure legends require more information to aid understanding.

I am a little confused about the aim of the study. According to the introduction, the purpose of generating the catanionic vesicles was for drug carrier and delivery. However, a large part of the experiments were focused on the characterisation of the antibacterial properties of the materials. It is not clear if the author also intended to use the catanionic vesicles as an alternative of the antibiotic medication. If so, will the vesicles be used as a type of antimicrobial medicine and/or sterilization solution? It needs some discussions to clarify the potential applications of the materials. The aim of the study needs to be much more focused.

We have included a new paragraph (Line 118) in the introduction to make clear which is the purpose of this work: “The study had three main aims: first, to prepare catanionic mixtures using green biocompatible surfactants, second, to design new formulations with high antimicrobial activities and moderate toxicity and third, to shed light on how the mixing ratio, the number of alkyl chains in the ion pairs and the nature of polar heads affect the biological and physico-chemical properties of catanionic vesicles. We expect that the findings described in this work will contribute to the understanding of the biological activity of catanionic mixtures from amino acid-based surfactants and help to rationalize the design of new and safe antimicrobial formulations”.

The results and discussions are all mixed up, which makes the section very lengthy. It is very difficult to pinpoint the actual results as they are embedded in the discussions. It would be beneficial to separate the discussion from the result to highlight the actual results.

To keep the length of results and discussion as short as possible, the corresponding section has been reorganized in subsections one for each studied subject. Then in each subsection we first describe the results and then we discuss them.

Minors:

Line 92, please check the “es” in “…value [72] es one or…” is correct.

Fixed

In Figure 1, (d) came before (c). It’s better to change their position.

Fixed (Figure 1)

Figure 4 needs to label A) and B)

Figure 4 is now Figure 3 and labels included (Figure 3)

The legends of Figure 8, 9, 10, and 13 need much more comprehensive.

The Figures has been renumbered. Figure 8 and 9 are now Figure 6, Figure 10 is Figure 7 and Figure 13 is Figure 10.

Legends have been rewritten (Line 543,  Line 635, Line 768)

Reviewer 4 Report

The manuscript, Biocompatible Catanionic Vesicles from Arginine-Based Surfactants: a New Strategy to Tune the Antimicrobial Activity and Cytotoxicity of Vesicular Systems, by Pinazo et al., has a merit to publish, however, there are major concerns to publish it in its current form. A lot of formatting issues are found in the tables, figures, or schemes.

  1. The authors should provide a reasonable explanation for the chosen combinations in the first paragraph of the results section.
  2. Almost EVERY FIGURE/scheme associated with certain type of errors or typos. It must be fixed it. Figure subpanels should require figure labels (A, B, C…)
  3. In Tables, some of the column heading are in bold and others not.
  4. In my opinion, at one point, it is hard to follow, too many abbreviations.
  5. According to IUPAC guidelines, zeta potential should be labelled as zeta potential or ζ-potential, not Z-potential.
  6. Several occasions, reference citations are needed. For example, in the introductory sentences of hemolytic activity.
  7. Hemolysis is misspelled in Figure 11.
  8. The title includes, Biocompatible Catanionic Vesicles from 2 Arginine-Based Surfactants. But the reported experiments did not support this. The authors only reported hemolytic activity, however, I recommend a couple of more experiments for instance cell compatibility, platelet activation, complement activation, or some coagulation experiments.

Author Response

Thank you very much for your constructive evaluation, which has helped us improve this manuscript. Below you will find our answers (in red) to your concerns (in black).

Comments and Suggestions for Authors

The manuscript, Biocompatible Catanionic Vesicles from Arginine-Based Surfactants: a New Strategy to Tune the Antimicrobial Activity and Cytotoxicity of Vesicular Systems, by Pinazo et al., has a merit to publish; however, there are major concerns to publish it in its current form. A lot of formatting issues are found in the tables, figures, or schemes.

 The authors should provide a reasonable explanation for the chosen combinations in the first paragraph of the results section.

Following the referee suggestion the next paragraph has been added at the first paragraph of the results and discussion section (Line 278-Line288): “In this work we used 2 cationic amino acid-based surfactants and 3 anionic surfactants (Figure 1) to prepare a range of catanionic mixtures at room temperature. The surfactant mixtures contained different proportions of: a) LAM and sodium myristate, both with a single chain (LM), b) double-chain cationic gemini C3(CA)2 and single-chain anionic sodium laurate (C3L) and c) C3(CA)2 and anionic 8 S-H, both with a double chain (C3S) (Figure 2). Spontaneous vesicle formation by binary mixtures of oppositely charged surfactants depends on both the composition of the formulation and the surfactant structure. Usually a single phase of catanionic vesicles is formed in the diluted cation- or anion-rich region of the surfactant [28]. Thus, in this work each mixture was prepared in 3 different proportions: one rich in the cationic surfactant, one rich in the anionic surfactant and an equimolecular formulation. Each system was prepared at two different total concentrations (1 and 5 mM)”

 Almost EVERY FIGURE/scheme associated with certain type of errors or typos. It must be fixed it. Figure subpanels should require figure labels (A, B, C…)

Tables, figures and schemes have been carefully revised and figure labels have been incorporated.

 In Tables, some of the column heading are in bold and others not.

The column headings in tables have been standardized

In my opinion, at one point, it is hard to follow, too many abbreviations.

I agree with the referee, this manuscript contains a lot of acronyms. However we think that their use is necessary to name the different formulations studied in this work. In this regards we included Table 1 indicating clearly the means of every abbreviations to make very easy to follow the results and discussion section.

According to IUPAC guidelines, zeta potential should be labelled as zeta potential or ζ-potential, not Z-potential.

z-potential has been changed by ζ-potential

Several occasions, reference citations are needed. For example, in the introductory sentences of hemolytic activity.

The Materials and Methods section in the original manuscript contained the Reference 27 (Line 251) (Pape, W. J.; Pfannenbecker, U.; Hoppe, U. Validation of the Red Blood Cell Test System as in Vitro Assay for the Rapid Screening of Irritation Potential of Surfactants. Mol. Toxicol. 1 (4), 525–536). Now we have also mentioned this reference in the introductory sentences of hemolytic activity (Line 666)

Hemolysis is misspelled in Figure 11.

Figure 11 (Now Figure 8) has been revised

The title includes, Biocompatible Catanionic Vesicles from 2 Arginine-Based Surfactants. But the reported experiments did not support this. The authors only reported hemolytic activity, however, I recommend a couple of more experiments for instance cell compatibility, platelet activation, complement activation, or some coagulation experiments.

This work is a multidisciplinary study that involves the physico-chemical and biological characterization of catanionic vesicles. In this regard, the red blood cells (RBCs) appear to be an excellent model to evaluate toxicity of molecules. The hemolytic activity of a compound is an indicator of its general cytotoxicity towards normal cells. RBCs are a good research model, because they do not have internal organelles, which make them an ideal cell system for studying basic drug–membrane interaction. Hemolysis is thus a simple, rapid, and useful prescreening technique to determine the surfactant biocompatibility. We know that a complete study of the biocompatibility of a drug would require bioassays using different human and animal cells. However, the use of different cell lines to determine the biocompatibility of these catanionic systems would have enlarged too much this manuscript.

Reviewer 5 Report

This is a nicely written manuscript with well-performed experiments. I have some comments:

  1. Figure 8 and 9 need to be presented more clearly. Antimicrobial activity is tested against ■ MRSA 551 ■ B. subtilis ■ K. rhizophila ■ S. epidermidis ■ K. pneumoniae ■ P. aeruginosa ■ E. coli ■ 552 C. albicans, but the figure does not mention which color/data correspond to which strain. Also, error bar is needed on each data set.
  2. DNA binding properties should preferentially be studied by gel electrophoresis using ethidium bromide stain. If quantifying using fluorescence emission spectra, please include standard deviation across at least 2 samples for each group.
  3. The manuscript is a bit lengthy. The authors should condense the introduction and some of the figures should be combined to save the journal space, e.g. Figures 3 & 4; 8 & 9; 6 & 7.

  4. Please correct the alignment of each column in Table 4.

  5. Figure markers a, b, etc are missing on  Figure 7 and Figure 6c.

Author Response

Thank you very much for your constructive evaluation, which has helped us improve this manuscript. Below you will find our answers (in red) to your concerns (in black).

Comments and Suggestions for Authors

This is a nicely written manuscript with well-performed experiments. I have some comments:

Figure 8 and 9 need to be presented more clearly. Antimicrobial activity is tested against ■ MRSA 551 ■ B. subtilis ■ K. rhizophila ■ S. epidermidis ■ K. pneumoniae ■ P. aeruginosa ■ E. coli ■ 552 C. albicans, but the figure does not mention which color/data correspond to which strain. Also, error bar is needed on each data set.

In the revised manuscript figure 8 and 9 has been merged in Figure 6 and the figure legend has been changed to clarify which color corresponds to every strain.

The minimum inhibitory concentration (MIC) values have been determined using the broth microdilution method. This procedure is carried out using two-fold dilutions of pure surfactants and their catanionic mixtures. This is a qualitative procedure to determine the antimicrobial activity because of that we have not include the error bars in this figure.

 DNA binding properties should preferentially be studied by gel electrophoresis using ethidium bromide stain. If quantifying using fluorescence emission spectra, please include standard deviation across at least 2 samples for each group.

I agree with the referee, gel electrophoresis using ethidium bromide is usually employed to determine DNA-surfactant interactions. However fluorescence measurement using ethidium bromide also gives useful information about the DNA binding capacity of surfactants. In this regards, previous studies showed us that the results obtained using gel electrophoresis agree with that achieved using ethidium bromide fluorescence measurements (Pinazo, A.; Pons, R.; Bustelo, M.; Manresa, M. Á.; Morán, C.; Raluy, M.; Pérez, L. Gemini Histidine Based Surfactants: Characterization; Surface Properties and Biological Activity. J. Mol. Liq. 2019, 289, 111156 . https://doi.org/10.1016/j.molliq.2019.111156.). Moreover standard deviations have been incorporated in Figure 10.

The manuscript is a bit lengthy. The authors should condense the introduction and some of the figures should be combined to save the journal space, e.g. Figures 3 & 4; 8 & 9; 6 & 7.

The introduction and some part of the result and discussion section have been reduced. Additionally, Figures 3 & 4; 8 & 9; 6 & 7 have been combined.

Please correct the alignment of each column in Table 4.

Table 4 has been corrected.

Figure markers a, b, etc are missing on  Figure 7 and Figure 6c.

The markers have been introduced in these figures (Figure 5 in the revised manuscript) .

Round 2

Reviewer 4 Report

As the authors addressed most of the concerns, I would go ahead with publishing this manuscript.